# Event-Triggered Finite-Time Formation Control of Underactuated Multiple ASVs with Prescribed Performance and Collision Avoidance

**DOI:** 10.3390/s23156756

**Published:** 2023-07-28

**Authors:** Xuehong Tian, Jianfei Lin, Haitao Liu, Xiuying Huang

**Affiliations:** 1Shenzhen Institute of Guangdong Ocean University, Shenzhen 518120, China; gdtianxh@126.com (X.T.); gdlinjf@126.com (J.L.); gdhuangxiuying@126.com (X.H.); 2School of Mechanical Engineering, Guangdong Ocean University, Zhanjiang 524088, China

**Keywords:** underactuated multiple ASVs, collision avoidance, finite-time control, event-triggered control, barrier Lyapunov function

## Abstract

In this paper, an event-triggered finite-time controller is proposed for solving the formation control problems of underactuated multiple autonomous surface vessels (ASVs), including asymmetric mass matrix, collision avoidance, maintaining communication distances and prescribed performance. First, to not only avoid collisions between the follower and leader but also maintain an effective communication distance, a desired tracking distance is designed to be maintained. Second, an improved barrier Lyapunov function (BLF) is proposed to implement the tracking error constraint. In addition, the relative threshold event-triggering strategy effectively solves the communication pressure problem and greatly saves communication resources. Finally, based on coordinate transformation, line of sight (LOS) and dynamic surface control (DSC), a comprehensive finite-time formation control method is proposed to avoid collisions and maintain communication distance. All the signals of the proposed control system can be stabilized in finite time (PFS). The numerical simulation results verify the effectiveness of the proposed control system.

## 1. Introduction

In recent years, due to the increasing demand for ocean exploration, autonomous surface vessels (ASVs) have been widely used in ocean exploration and exploitation and have played an increasingly important role [1]. However, it has been quite difficult for single autonomous surface vessels (ASVs) to accomplish some exploration tasks, such as large-scale cruising, the formation of combat formations and complex sea exploration. Therefore, cooperative control of multiple ASVs is necessary and meaningful [2]. Among various cooperative control methods, formation control is widely applied due to its simple structure and scalability [3,4,5], so formation control has received much attention in ASV motion control [6]. However, most multiple ASV formation control is used for fully actuated ASVs, but an underactuated system can not only reduce the system cost and allow for a simpler structure but also provide an emergency control strategy in the case of an actuator failure of the fully driven system. Therefore, the formation control of underactuated ASVs is very worthy of attention.

Recently, due to the complexity of the task performed by the ASV, the requirements for the motion performance of the ASV have increased. In [7], an adaptive neural network trajectory tracking controller with an output saturation model is proposed. To improve the performance of the control system, a barrier Lyapunov function is introduced to achieve the prescribed performance. To handle the angle and LOS range constraints, the BLF was introduced into the control scheme [8]. In [9], an output feedback controller was designed using a log-type BLF to solve the output constraint problem. Based on a neural network observer, tan-type BLF and DSC technique, an adaptive controller was proposed in [10] that not only handles output constraints but also avoids collisions and maintains ASV connectivity. An improved BLF was proposed in the control strategy to solve the error constraint in [11], which is applicable both with and without constraints. However, in the practical ocean engineering environment, communication between ASVs and sensors is usually limited [12]. Therefore, communication between ASVs is not always maintained, thereby raising connectivity problems [13]. In addition, collision avoidance of ASVs is an important problem to be considered in leader–follower formation strategies [14]. In [15], a new formation error based on nonlinear transformation was proposed to realize initial connectivity protection and collision avoidance. By introducing a performance function and BLF in [16], communication connectivity is maintained, collisions between the formation vehicles are avoided, and the tracking error reaches the specified performance. Avoiding obstacles while avoiding collisions and maintaining connectivity of ASVs was achieved in [17], which was further advanced for collision avoidance studies. In short, the introduction of a BLF can effectively improve the system performance and avoid collisions. However, for underactuated ASVs, among existing BLFs, the use of log-type BLFs is limited by performance functions, while tan-type BLFs add additional complexity to controller design [12].

In practice, it is not only limited by the effective sensing range of the sensor and the communication distance of the communication equipment; the communication bandwidth is often limited as well [18]. In [19], an adaptive controller based on event triggering was proposed by combining the event-triggered strategy with the adaptive law, which does not need to obtain the current ASV state at any time and reduces the communication resources needed. For cooperative estimation with communication delay, an event-triggered delay-based distributed state observer was designed in [20]. A sliding mode control scheme based on an event-triggered strategy was proposed in [21], and this control method can effectively reduce communication bandwidth compared to conventional controllers without event-triggered methods. The above event-triggered strategies are all based on a fixed threshold to determine whether the triggering conditions are satisfied, which can make the event-triggered controller trigger frequently before the system stabilizes in response to large changes in control inputs.

In the above work, the ASV tracking errors are asymptotically convergent, which means that the error converges to the origin exponentially. In fact, the stable control of formation always needs to be completed in a finite time to improve the stable speed of the formation control system [22]. In [23], a finite-time control method was developed for ASVs with error constraints by introducing a BLF and saturation functions. A finite-time formation controller based on a neural network is proposed to solve the actuator faults and unknown dynamics in multiple underactuated ASVs in [24]. However, none of the above works consider collision avoidance or communication bandwidth issues.

Motivated by the above discussion, and considering the nondiagonal inertia matrix, a leader–follower finite-time formation control system based on an event-triggered strategy is proposed to achieve the maintenance of communication effectiveness and collision avoidance. In this work, a BLF is introduced to maintain communication effectiveness and collision avoidance. To avoid the employment of derivatives of the virtual control signals, a DSC technique is employed. Limited by the communication bandwidth, an event-triggered strategy based on a relative threshold is proposed. Therefore, the main contributions are as follows:(1)An improved BLF is developed to guarantee both prescribed transient error tracking and steady performance. Differently from the existing works in [9,23], maintaining communication distance and collision avoidance between ASVs is also achieved.(2)Compared with existing works [25], a relative threshold-based event-triggered controller is proposed to reduce the communication bandwidth. Compared with the existing fixed threshold event-triggered controllers [26,27], the relative threshold strategy can achieve fewer stable errors.(3)A finite-time event-triggered formation tracking control strategy is proposed to solve the error constraint problem of underactuated multi-ASV formation. In the control system, all signals are practical finite-time-stable (PFS), which is different from the existing works on ASV tracking control with constraints [11,12,28].

The rest of the paper is organized as follows. Section 2 describes the preliminaries and problem formulation. Section 3 describes the formation controller design. Section 4 describes the simulations. Finally, Section 5 gives the conclusions.

## 2. Preliminaries and Problem Formulation

### 2.1. Preliminaries

**Lemma** **1**[29]. *Define a system* $\dot{x}=f(x)$*, if* $\kappa_{1}>0$*,* $\kappa_{2}>0$*,* $\varepsilon_{l}>0$ *and* β
∈(0,1) *such that*


(1)
$$\dot{V}(x)\leq -\kappa_{1}V(x)-\kappa_{2}V^{\beta }(x)+\varepsilon_{l}$$



*Therefore, the continuous nonlinear system is a practical finite-time-stable (PFS) system with the residual set *

$\Omega_{L}=\min\left\{\frac{\varepsilon_{l}}{(1-H)\kappa_{1}},{\left(\frac{\varepsilon_{l}}{(1-H)\kappa_{2}}\right)}^{1/\beta }\right\}$

* of the solution, where *

0<H<1

*. The settling time is:*

(2)
$$T(x_{0})\leq \max\left\{\frac{1}{\kappa_{1}H(1-\beta )}\ln\left(\frac{\kappa_{1}HV^{1-\beta }(x_{0})+\kappa_{2}}{\kappa_{2}}\right),\frac{1}{\kappa_{1}(1-\beta )}\ln\left(\frac{\kappa_{1}V^{1-\beta }(x_{0})+\kappa_{2}H}{\kappa_{2}H}\right)\right\}$$



**Lemma** **2**[30]. *For a∈ℝ*
*and* $\varepsilon_{0}>0$ *, the following inequality holds:*


(3)
$$0\leq \left|a\right|-a\tanh\left(\frac{a}{\varepsilon_{0}}\right)\leq 0.2785\varepsilon_{0}$$


### 2.2. Model of Underactuated ASVs

The modeling of the i-th underactuated ASV is as follows [31]:(4)$$\left\{\begin{array}{l} {\dot{\eta }}_{i}=J_{i}(\psi_{i})\upsilon_{i}\\ {\dot{\upsilon }}_{i}=M_{i}^{-1}(-C_{i}(\upsilon_{i})-D_{i}(\upsilon_{i})+\tau_{i}) \end{array}\right.,\;i=1,2,\ldots ,n$$
where
(5)$$\begin{array}{cc} J_{i}(\psi_{i})=\left[\begin{array}{ccc} \cos\psi_{i} & -\sin\psi_{i} & 0\\ \sin\psi_{i} & \cos\psi_{i} & 0\\ 0 & 0 & 1 \end{array}\right], & M_{i}=\left[\begin{array}{ccc} m_{i,11} & 0 & 0\\ 0 & m_{i,22} & m_{i,23}\\ 0 & m_{i,32} & m_{i,33} \end{array}\right],\\ D_{i}(\upsilon_{i})=\left[\begin{array}{ccc} d_{i,11} & 0 & 0\\ 0 & d_{i,22} & d_{i,23}\\ 0 & d_{i,32} & d_{i,33} \end{array}\right], & C_{i}(\upsilon_{i})=\left[\begin{array}{ccc} 0 & 0 & C_{i,13}\\ 0 & 0 & C_{i,23}\\ C_{i,31} & C_{i,32} & 0 \end{array}\right] \end{array}$$

In this paper, $M_{i}$, $C_{i}(\upsilon_{i})$ and $D_{i}(\upsilon_{i})$ denote the mass matrix, the total Coriolis and the centripetal acceleration matrix, respectively. $J_{i}(\upsilon_{i})$ is a rotation matrix. $\eta_{i}={\left[x_{i},y_{i},\psi_{i}\right]}^{T}$ is the position. $\left(x_{i},y_{i}\right)$ and $\psi_{i}$ denote the position and yaw angle, respectively. $\tau_{ci}={\left[\tau_{ui},0,\tau_{ri}\right]}^{T}$ is the event-triggered control input of the ASV. $\upsilon_{i}={\left[u_{i},v_{i},r_{i}\right]}^{T}$ is the velocity vector. The geometric structure is shown in Figure 1. $\omega_{i}$ and $\phi_{i}$ are the LOS range and angle, respectively.

In System (4), both the yaw angle and yaw velocity are affected by the control input $\tau_{ri}$ due to the mass matrix $M_{i}$. Therefore, the coordinate transformation is described as [6]:(6)$$\left\{\begin{array}{l} {\bar{x}}_{i}=x_{i}+\varsigma_{i}\cos\psi_{i}\\ {\bar{y}}_{i}=y_{i}+\varsigma_{i}\sin\psi_{i}\\ {\bar{v}}_{i}=v_{i}+\varsigma_{i}r_{i} \end{array}\right.$$
where $\varsigma_{i}=\frac{m_{i,23}}{m_{i,22}}$. With Equation (6), System (4) can be rewritten as:(7)$$\left\{\begin{array}{l} {\dot{\bar{\eta }}}_{i}=J_{i}(\psi_{i}){\bar{\upsilon }}_{i}\\ {\dot{\bar{\upsilon }}}_{i}=f_{i}+\delta_{i}{}^{T}\tau_{ci} \end{array}\right.$$
where ${\bar{\eta }}_{i}={\left[{\bar{x}}_{i},{\bar{y}}_{i},\psi_{i}\right]}^{T}$, ${\bar{\upsilon }}_{i}={\left[u_{i},{\bar{v}}_{i},r_{i}\right]}^{T}$, $f_{i}={\left[f_{i1},f_{i2},f_{i3}\right]}^{T}$, $\delta_{i}={\left[1/m_{i,11},0,m_{i,22}/{\bar{m}}_{i,33}\right]}^{T}$ and ${\bar{m}}_{i,33}=m_{i,22}m_{i,33}-m_{i,23}^{2}$. The vectors $f_{i}$ are defined as:(8)$$\left\{\begin{array}{l} f_{i1}=\left(m_{i,22}v_{i}r_{i}+m_{i,23}r_{i}^{2}-d_{i,11}u_{i}\right)/m_{i,11}\\ f_{i2}=\left(-m_{i,11}u_{i}r_{i}-d_{i,22}v_{i}-d_{i,23}r_{i}\right)/m_{i,22}\\ f_{i3}=\left((m_{i,11}m_{i,22}-m_{i,22}^{2})u_{i}v_{r}+(m_{i,11}m_{i,32}-m_{i,23}m_{i,22})u_{i}v_{r}-m_{i,22}(d_{i,33}r_{i}+d_{i,32}v_{i})+m_{i,23}(d_{i,23}r_{i}+d_{i,22}v_{i})\right)/\left(m_{i,22}m_{i,33}-m_{i,23}m_{i,32}\right) \end{array}\right.$$

### 2.3. Leader–Follower Formation Architecture

In this paper, a formation tracking controller with the desired distance is designed with a leader–follower architecture as the objective. Figure 2 shows the tracking relationship and communication topology between the leader and follower.

The LOS is introduced to facilitate the subsequent control design. For any group of leader–follower formations, i is defined as the follower number and i−1 as the leader number. The LOS range $\omega_{i}$ and angle $\phi_{i}$ are defined as
(9)$$\left\{\begin{array}{l} \omega_{i}(t)=\sqrt{{\left({\bar{x}}_{i-1}-{\bar{x}}_{i}\right)}^{2}+{\left({\bar{y}}_{i-1}-{\bar{y}}_{i}\right)}^{2}}\\ \phi_{i}(t)=\mathrm{atan}2\left({\bar{y}}_{ei},{\bar{x}}_{ei}\right) \end{array}\right.$$
and
(10)$$\left[\begin{array}{c} {\bar{x}}_{ei}\\ {\bar{y}}_{ei} \end{array}\right]=\left[\begin{array}{cc} \cos\psi_{i} & \sin\psi_{i}\\ -\sin\psi_{i} & \cos\psi_{i} \end{array}\right]\left[\begin{array}{c} {\bar{x}}_{i-1}-{\bar{x}}_{i}\\ {\bar{y}}_{i-1}-{\bar{y}}_{i} \end{array}\right]$$
(11)$$\mathrm{atan}2({\bar{y}}_{ei},{\bar{x}}_{ei})=\left\{\begin{array}{ll} \arctan\left(\frac{{\bar{y}}_{ei}}{{\bar{x}}_{ei}}\right) & \mathrm{if}\;{\bar{x}}_{ei}>0,\\ \arctan\left(\frac{{\bar{y}}_{ei}}{{\bar{x}}_{ei}}\right)+\pi & \mathrm{if}\;{\bar{x}}_{ei}<0\;\mathrm{and}\;{\bar{y}}_{ei}\geq 0,\\ \arctan\left(\frac{{\bar{y}}_{ei}}{{\bar{x}}_{ei}}\right)-\pi & \mathrm{if}\;{\bar{x}}_{ei}<0\;\mathrm{and}\;{\bar{y}}_{ei}<0,\\ +\frac{\pi }{2} & \mathrm{if}\;{\bar{x}}_{ei}=0\;\mathrm{and}\;{\bar{y}}_{ei}>0,\\ -\frac{\pi }{2} & \mathrm{if}\;{\bar{x}}_{ei}=0\;\mathrm{and}\;{\bar{y}}_{ei}<0,\\ \mathrm{undefined} & \mathrm{if}\;{\bar{x}}_{ei}=0\;\mathrm{and}\;{\bar{y}}_{ei}=0, \end{array}\right.$$

Define ${\bar{\varpi }}_{i}$ as the maximum distance designed, depending on the measuring capability of the sensor. ${\underline{\varpi }}_{i}$ denotes the minimum safe distance. ${\underline{\phi }}_{i}$ and ${\bar{\phi }}_{i}$ represent the minimum and maximum of the angle detected by the sensor, respectively. $\varpi_{i}$ and $\phi_{i}$ are described by:(12)$$\left\{\begin{array}{c} {\underline{\varpi }}_{i}<\omega_{i}(t)<{\bar{\varpi }}_{i}\\ {\underline{\phi }}_{i}<\phi_{i}(t)<{\bar{\phi }}_{i} \end{array}\right.$$

Based on (12), $\varpi_{i,des}$ and $\phi_{i,des}$ represent the desired distance and angle, respectively, which can avoid collisions and maintain communication. Therefore, we define the tracking errors as follows:(13)$$\left\{\begin{array}{c} e_{\varpi i}(t)=\omega_{i}-\varpi_{i,des}\\ e_{\phi i}(t)=\phi_{i}-\phi_{i,des} \end{array}\right.$$
where $\varpi_{i,des}=\left({\underline{\varpi }}_{i}+{\bar{\varpi }}_{i}\right)/2$ and $\phi_{i,des}=\left({\underline{\phi }}_{i}+{\bar{\phi }}_{i}\right)/2$. From (12) and (13), $e_{\varpi i}$ and $e_{\phi i}$ satisfy:(14)$$\left\{\begin{array}{c} {\underline{\varpi }}_{i}-\varpi_{i,des}<e_{\varpi i}(t)<{\bar{\varpi }}_{i}-\varpi_{i,des}\\ {\underline{\phi }}_{i}-\phi_{i,des}<e_{\phi i}(t)<{\bar{\phi }}_{i}-\phi_{i,des} \end{array}\right.$$

In this work, a performance function is incorporated to guarantee the performance of formation control. Thus, the errors satisfy the following inequalities:(15)$$\left\{\begin{array}{c} -L_{\varpi i}(t)<e_{\varpi i}(t)<L_{\varpi i}(t)\\ -L_{\phi i}(t)<e_{\phi i}(t)<L_{\phi i}(t) \end{array}\right.$$

In addition, the performance function is designed as:(16)$$\left\{\begin{array}{c} L_{\varpi i}(t)=(L_{\varpi i,0}-L_{\varpi i,\infty })\exp(-\beta_{\varpi i}t)+L_{\varpi i,\infty }\\ L_{\phi i}(t)=(L_{\phi i,0}-L_{\phi i,\infty })\exp(-\beta_{\phi i}t)+L_{\phi i,\infty } \end{array}\right.$$
where $L_{\varpi i,0}={\bar{\varpi }}_{i}-\varpi_{i,des}$ and $L_{\phi i,0}={\bar{\phi }}_{i}-\phi_{i,des}$. $\beta_{\varpi i}$ and $\beta_{\phi i}$ denote the convergence speed. $L_{\varpi i,\infty }$ and $L_{\phi i,\infty }$ represent the maximum values after the error has stabilized. If the designed control law makes Equations (15) and (16) hold, the error constraint of (14) is satisfied, which means that the Inequality (12) holds.

**Assumption 1:** *The desired trajectories* $h_{0}$ *and* ${\dot{h}}_{0}$ *are bounded.*

**Remark** **1:***From (13), the tracking error performance is consistent with the distance error performance. When the distance error* $\omega_{i}$ *converges to near the desired distance, the tracking error* $e_{\varpi i}$ *converges to a small region near zero.*

**Remark** **2:***When achieving collisions and effective communication distance, Inequality (12) can be held to satisfy* $0<{\underline{\omega }}_{i}<\omega_{i}<{\bar{\omega }}_{i}$ *so that* ${\bar{x}}_{ei}=0$ *and* ${\bar{y}}_{ei}=0$ *do not hold simultaneously, which avoids the undefined point of* $\phi_{i}$. *Even if there is some measurement error in the onboard sensors, the ASV formation will work equally well as long as the distance between the ASVs remains within the communication range.*

## 3. Formation Controller Design

In this section, a modified BLF is designed for the controller to meet the constraints of the formation tracking error. Then, a finite-time formation controller is proposed, which combines DSC technology and an event-triggering mechanism. Finally, the system’s stability is proven.

### 3.1. Barrier Lyapunov Function

A BLF is developed as follows:(17)$$\left\{\begin{array}{l} V_{\varpi i}=\frac{1}{2}{\left(\frac{L_{\varpi i}e_{\varpi i}}{L_{\varpi i}-e_{\varpi i}}\right)}^{2}\\ V_{\phi i}=\frac{1}{2}{\left(\frac{L_{\phi i}e_{\phi i}}{L_{\phi i}-e_{\phi i}}\right)}^{2} \end{array}\right.$$

Even if there is no constraint on the formation tracking error, the following can be obtained:(18)$$\left\{\begin{array}{l} \underset{L_{\varpi i}\to \infty }{\lim}\frac{1}{2}{\left(\frac{L_{\varpi i}e_{\varpi i}}{L_{\varpi i}-e_{\varpi i}}\right)}^{2}=\frac{e_{\varpi i}^{2}}{2}\\ \underset{L_{\phi i}\to \infty }{\lim}\frac{1}{2}{\left(\frac{L_{\phi i}e_{\phi i}}{L_{\phi i}-e_{\phi i}}\right)}^{2}=\frac{e_{\phi i}^{2}}{2} \end{array}\right.$$

This means that this BLF can be discussed as a special case of constraint requirements on systems with no constraint requirements. Therefore, Equation (17) is a general BLF, which can be regarded as an unconstrained universal BLF.

**Remark** **3:***Obviously,* $V_{ni}=0$ *if and only if* $e_{ni}=0$*, and the minimum value of* $e_{ni}$ *is 0. When* $e_{ni}\to L_{ni}$*, there exists* $V_{ni}\to +\infty$*. This shows that the errors* $e_{ni}$ *will not exceed* $L_{ni}$ *as long as* $V_{ni}$ *is bounded, and* n=ϖ,ϕ*.*

**Remark** **4:***The log-type BLF methods are also used in the underactuated ASV control strategy [9,28];* $V_{ni}=\frac{1}{2}\log\left(\frac{L_{ni}^{2}}{L_{ni}^{2}-e_{ni}^{2}}\right),n=\varpi ,\phi$*. When* $L_{ni}\to +\infty$*,* $V_{ni}=0$*, there is no limit to the errors. Thus, the log-type BLF cannot be regarded as a universal BLF in unconstrained conditions. It is worth mentioning that a tan-type BLF can be implemented with or without constraint situations [12,32]. Unfortunately, a tan-type BLF adds complexity to the controller. However, the BLF (17) developed has a simple structure and is suitable for both constrained and unconstrained cases.*

### 3.2. Finite-Time Formation Controller Design

In this work, the formation controller design includes two steps, namely, kinematic controller design and dynamic controller design.

Step 1: The errors are given by:(19)$$\left\{\begin{array}{l} e_{ui}=u_{i}-\alpha_{f,ui}\\ e_{ri}=r_{i}-\alpha_{f,ri} \end{array}\right.$$
The boundary layer errors are defined as follows:(20)$$\left\{\begin{array}{l} e_{f,ui}=\alpha_{f,ui}-\alpha_{ui}\\ e_{f,ri}=\alpha_{f,ri}-\alpha_{ri} \end{array}\right.$$
where $\alpha_{ui}$ is the virtual control of u and $\alpha_{ri}$ is the virtual control of r. $\alpha_{f,ui}$ and $\alpha_{f,ri}$ are the filtered inputs obtained from Filter (26), respectively. Consider Equation (13), whose derivative along System (9) is:(21)$$\left\{\begin{array}{l} {\dot{e}}_{\varpi i}=-u_{i}\cos\phi_{i}-{\bar{v}}_{i}\sin\phi_{i}+{\bar{x}}_{i-1}\cos(\psi_{i}+\phi_{i})+{\bar{y}}_{i-1}\sin(\psi_{i}+\phi_{i})\\ {\dot{e}}_{\phi i}=\left[u_{i}\sin\phi_{i}-{\bar{v}}_{i}\cos\phi_{i}-{\bar{x}}_{i-1}\sin(\psi_{i}+\phi_{i})+{\bar{y}}_{i-1}\cos(\psi_{i}+\phi_{i})\right]/\varpi_{i}-r_{i} \end{array}\right.$$

Consider the Lyapunov function candidate $V_{i1}$
(22)$$V_{i1}=\frac{1}{2}{\left(\frac{L_{\varpi i}e_{\varpi i}}{L_{\varpi i}-e_{\varpi i}}\right)}^{2}+\frac{1}{2}{\left(\frac{L_{\phi i}e_{\phi i}}{L_{\phi i}-e_{\phi i}}\right)}^{2}$$
Its derivative is:(23)$${\dot{V}}_{i1}=\frac{L_{\varpi i}e_{\varpi i}}{L_{\varpi i}-e_{\varpi i}}\left(\frac{L_{\varpi i}^{2}{\dot{e}}_{\varpi i}-{\dot{L}}_{\varpi i}e_{\varpi i}^{2}}{{\left(L_{\varpi i}-e_{\varpi i}\right)}^{2}}\right)+\frac{L_{\phi i}e_{\phi i}}{L_{\phi i}-e_{\phi i}}\left(\frac{L_{\phi i}^{2}{\dot{e}}_{\phi i}-{\dot{L}}_{\phi i}e_{\phi i}^{2}}{{\left(L_{\phi i}-e_{\phi i}\right)}^{2}}\right)$$

For Equation (23) and velocity errors described by (19), (20) and (21), the virtual controller can be designed as:(24)$$\alpha_{ui}=\frac{1}{\cos\phi_{i}}\left[-{\bar{v}}_{i}\sin\phi_{i}+{\dot{q}}_{i-1}P_{i1}+k_{di}e_{\varpi i}\frac{\left(L_{\varpi i}-e_{\varpi i}\right)}{L_{\varpi i}}-\frac{{\dot{L}}_{\varpi i}}{L_{\varpi i}^{2}}e_{\varpi i}^{2}+L_{di}L_{\varpi i}^{-3/2}e_{\varpi i}^{1/2}{\left(L_{\varpi i}-e_{\varpi i}\right)}^{3/2}\right]$$
(25)$$\alpha_{ri}=\frac{u_{i}\sin\phi_{i}-{\bar{v}}_{i}\cos\phi_{i}+{\dot{q}}_{i-1}P_{i2}}{\varpi_{i}}+k_{ai}e_{\phi i}\frac{\left(L_{\phi i}-e_{\phi i}\right)}{L_{\phi i}}-\frac{{\dot{L}}_{\phi i}}{L_{\phi i}^{2}}e_{\phi i}^{2}+L_{ai}L_{\phi i}^{-3/2}e_{\phi i}^{1/2}{\left(L_{\phi i}-e_{\phi i}\right)}^{3/2}$$
with $P_{i1}={\left[\cos(\psi_{i}+\phi_{i}),\sin(\psi_{i}+\phi_{i})\right]}^{T}$, $P_{i2}={\left[-\sin(\psi_{i}+\phi_{i}),\cos(\psi_{i}+\phi_{i})\right]}^{T}$ and ${\dot{q}}_{i-1}=\left[{\dot{\bar{x}}}_{i-1},{\dot{\bar{y}}}_{i-1}\right]$. $k_{di}$, $k_{ai}$, $L_{di}$ and $L_{ai}$ are positive parameters.

**Remark** **5:***If* $\phi_{i}=\pm \frac{\pi }{2}$*, the virtual controller described by (24) is singular, but this can be avoided by (12). Therefore, we define* $\left|{\underline{\phi }}_{i}\right|<\frac{\pi }{2},\left|{\bar{\phi }}_{i}\right|<\frac{\pi }{2}$*.*

To avoid the “differential explosion” problem caused by the differential of the virtual signal, the DSC method is introduced in [33]. Thus, the first-order filter can be defined as:(26)$$\left\{\begin{array}{l} \xi_{ui}{\dot{\alpha }}_{f,ui}+\alpha_{f,ui}=\alpha_{mui},\alpha_{f,ui}(0)=\alpha_{mui}(0)\\ \xi_{ri}{\dot{\alpha }}_{f,ri}+\alpha_{f,ri}=\alpha_{mri},\alpha_{f,ri}(0)=\alpha_{mri}(0) \end{array}\right.$$
with $\alpha_{mui}=\alpha_{ui}+\xi_{ui}e_{\varpi i}s_{\varpi i}\cos\phi_{i}$ and $\alpha_{mri}=\alpha_{ri}+\xi_{ri}e_{\phi i}s_{\phi i}$, where $s_{\varpi i}=\frac{L_{\varpi i}^{3}}{{\left(L_{\varpi i}-e_{\varpi i}\right)}^{3}}$ and $s_{\phi i}=\frac{L_{\phi i}^{3}}{{\left(L_{\phi i}-e_{\phi i}\right)}^{3}}$. $\xi_{ui}$ and $\xi_{ri}$ are the filter time parameters. Then, the derivatives of $e_{f,ui}$ and $e_{f,ri}$ are:(27)$$\left\{\begin{array}{l} {\dot{e}}_{f,ui}=-\frac{e_{f,ui}}{\xi_{ui}}+e_{\varpi i}s_{\varpi i}\cos\phi_{i}-N_{ui}(\cdot )\\ {\dot{e}}_{f,ri}=-\frac{e_{f,ri}}{\xi_{ri}}+e_{\phi i}s_{\phi i}-N_{ri}(\cdot ) \end{array}\right.$$
where ${\dot{\alpha }}_{ui}≜N_{ui}(\cdot )$ and ${\dot{\alpha }}_{ri}≜N_{ri}(\cdot )$ with $N_{ui}(\eta_{i-1},{\dot{\eta }}_{i-1},{\ddot{\eta }}_{i-1},{\bar{\upsilon }}_{i},k_{\varpi i},{\dot{k}}_{\varpi i},{\ddot{k}}_{\varpi i},e_{\varpi i},e_{ui},e_{ri},e_{f,ui},e_{f,ri})$ and $N_{ri}(\eta_{i-1},{\dot{\eta }}_{i-1},{\ddot{\eta }}_{i-1},{\bar{\upsilon }}_{i},k_{\phi i},{\dot{k}}_{\phi i},{\ddot{k}}_{\phi i},e_{\phi i},e_{ui},e_{ri},e_{f,ui},e_{f,ri})$ being unknown continuous functions.

Substituting (19), (20), (21), (24) and (25) into (23) yields:(28)$$\begin{array}{ll} {\dot{V}}_{i1}= & -k_{di}{\left(\frac{L_{\varpi i}e_{\varpi i}}{L_{\varpi i}-e_{\varpi i}}\right)}^{2}-L_{di}{\left(\frac{L_{\varpi i}e_{\varpi i}}{L_{\varpi i}-e_{\varpi i}}\right)}^{3/2}-e_{ui}e_{\varpi i}s_{\varpi i}\cos\phi_{i}-e_{f,ui}e_{\varpi i}s_{\varpi i}\cos\phi_{i}\\ & -k_{ai}{\left(\frac{L_{\phi i}e_{\phi i}}{L_{\phi i}-e_{\phi i}}\right)}^{2}-L_{ai}{\left(\frac{L_{\phi i}e_{\phi i}}{L_{\phi i}-e_{\phi i}}\right)}^{3/2}-e_{ri}e_{\phi i}s_{\phi i}-e_{f,ri}e_{\phi i}s_{\phi i} \end{array}$$

Define the following Lyapunov function:(29)$$V_{i2}=V_{i1}+\frac{e_{f,ui}^{2}}{2}+\frac{e_{f,ri}^{2}}{2}$$
Its derivative along (20), (26), (27) and (28) is:(30)$$\begin{array}{c} {\dot{V}}_{i2}=-k_{di}{\left(\frac{L_{\varpi i}e_{\varpi i}}{L_{\varpi i}-e_{\varpi i}}\right)}^{2}-L_{di}{\left(\frac{L_{\varpi i}e_{\varpi i}}{L_{\varpi i}-e_{\varpi i}}\right)}^{3/2}-\frac{e_{f,ui}^{2}}{\xi_{ui}}-e_{ui}e_{\varpi i}s_{\varpi i}\cos\phi_{i}-e_{f,ui}N_{ui}(\cdot )\\ -k_{ai}{\left(\frac{L_{\phi i}e_{\phi i}}{L_{\phi i}-e_{\phi i}}\right)}^{2}-L_{ai}{\left(\frac{L_{\phi i}e_{\phi i}}{L_{\phi i}-e_{\phi i}}\right)}^{3/2}-\frac{e_{f,ri}^{2}}{\xi_{ri}}-e_{ri}e_{\phi i}s_{\phi i}-e_{f,ri}N_{ri}(\cdot ) \end{array}$$

Step 2. For System (7), the derivatives of (19) are as follows:(31)$$\left\{\begin{array}{l} {\dot{e}}_{ui}=f_{i1}+\frac{\tau_{ui}}{m_{i,11}}+\frac{e_{f,ui}}{\xi_{ui}}-e_{\varpi i}s_{\varpi i}\cos\phi_{i}\\ {\dot{e}}_{ri}=f_{i3}+\frac{m_{i,22}\tau_{ri}}{{\bar{m}}_{i,33}}+\frac{e_{f,ri}}{\xi_{ri}}-e_{\phi i}s_{\phi i} \end{array}\right.$$

Therefore, the actual controllers are designed as follows:(32)α2,ui=mi,11(−kuieui−fi1−ef,uiξui+2eϖisϖicosϕi−luieui12−Luief,ui32eui−1)α2,ri=m¯i,33mi,22(−krieri−fi3−ef,riξri+2eϖisϖi−lrieri12−Lrief,ri32eri−1)
where kui, kri, lui, lri, Lui and Lri are positive design parameters. For (32), the relative threshold event-triggered mechanism is further considered:(33)hui(t)=−(1+εui)α2,uitanheuiα2,uiσ+p¯uitanheuip¯uiσhri(t)=−(1+εri)α2,ritanheriα2,riσ+p¯ritanherip¯riσ
(34)τui=hui(tkui) ∀t∈tkui,tk+1uiτri=hri(tkri) ∀t∈tkri,tk+1ri
(35)tk+1ui=inft∈ℝ| Eui(t)≥εuiτui(t)+puitk+1ri=inft∈ℝ| Eri(t)≥εriτri(t)+pri
where Eui=hui−τui and Eri=hri−τri represent the measurement errors. εui, εri, σ, pui, pri, p¯ui and p¯ri are positive, with pui1−εui<p¯ui and pri1−εri<p¯ri. tu,k and tr,k, k∈ℤ+ denote the update times. The control laws in (34) are changed to hui(tk+1ui) and hri(tk+1ri), which indicates that the control inputs do not change at the time intervals t∈tkui,tk+1ui and t∈tkri,tk+1ri.

In this paper, an underactuated ASV formation controller based on finite-time theory, a BLF and event triggering is proposed. The proposed control system avoids zero behavior by adjusting the appropriate parameters, and the stability of the control system is demonstrated. The following theory is proposed in this work.

**Theorem** **1.***For ASV System (4), and under Assumption 1, consider the actual controller shown in (33)–(35) with the virtual control laws in (24) and (25). If given* $B_{ei}>0$*, the initial conditions satisfy* $V_{i3}(0)\leq B_{ei}/2$*. There exist design parameters* $\varepsilon_{ui}$*,* $\varepsilon_{ri}$*,* ${\bar{p}}_{ui}$*,* ${\bar{p}}_{ri}$*,* $p_{ui}$*,* $p_{ri}$*,* $\xi_{ui}$*,* $\xi_{ri}$*,* $k_{ui}$*,* $k_{ri}$*,* $k_{di}$*,* $k_{ai}$*,* $L_{di}$*,* $L_{ai}$*,* $l_{ui}$*,* $l_{ri}$*,* $L_{ui}$*,* $L_{ri}$*, such that* ${\dot{V}}_{i3}(x)\leq -\kappa_{i1}V_{i3}-\kappa_{i2}V_{i3}^{3/4}+\varepsilon_{i}^{*}$ *and:*

*(1)* 
*All signals of the control system are finite-time stable, and satisfying the tracking error constraint in (14) means that Inequality (12) also holds, which realizes collision avoidance and communication distance maintenance.*
*(2)* *There are times* $t_{ui}^{*}>0$*, the lower bound of the trigger interval* $t_{k+1}^{ui}-t_{k}^{ui}$ *is* $t_{ui}^{*}$*, and* $t_{k+1}^{ri}-t_{k}^{ri}$ *is* $t_{ri}^{*}$*, which means that there is no Zeno behavior in the proposed control system.*

**Proof.** Define the following Lyapunov function:

(36)$$V_{i3}=V_{i2}+\frac{e_{ui}^{2}}{2}+\frac{e_{ri}^{2}}{2}$$
Its derivative along (30) and (31) is:(37)$$\begin{array}{c} {\dot{V}}_{i3}=-k_{di}{\left(\frac{L_{\varpi i}e_{\varpi i}}{L_{\varpi i}-e_{\varpi i}}\right)}^{2}-L_{di}{\left(\frac{L_{\varpi i}e_{\varpi i}}{L_{\varpi i}-e_{\varpi i}}\right)}^{3/2}-\frac{e_{f,ui}^{2}}{\xi_{ui}}-2e_{ui}e_{\varpi i}s_{\varpi i}\cos\phi_{i}-e_{f,ui}N_{ui}(\cdot )+e_{ui}f_{i1}+\frac{\tau_{ui}e_{ui}}{m_{i,11}}+\frac{e_{f,ui}e_{ui}}{\xi_{ui}}\\ -k_{ai}{\left(\frac{L_{\phi i}e_{\phi i}}{L_{\phi i}-e_{\phi i}}\right)}^{2}-L_{ai}{\left(\frac{L_{\phi i}e_{\phi i}}{L_{\phi i}-e_{\phi i}}\right)}^{3/2}-\frac{e_{f,ri}^{2}}{\xi_{ri}}-2e_{ri}e_{\phi i}s_{\phi i}-e_{f,ri}N_{ri}(\cdot )+e_{ri}f_{i3}+\frac{m_{i,22}\tau_{ri}e_{ri}}{{\bar{m}}_{i,33}}+\frac{e_{f,ri}e_{ri}}{\xi_{ri}} \end{array}$$

From (35), in the intervals $\left[t_{k}^{ui},\right.\left.t_{k+1}^{ui}\right)$ and $\left[t_{k}^{ri},\right.\left.t_{k+1}^{ri}\right)$, we have
(38)$$\left\{\begin{array}{l} h_{ui}(t)=(1+\chi_{i1}(t)\varepsilon_{ui})\tau_{ui}(t)+\chi_{i2}(t)p_{ui}\\ h_{ri}(t)=(1+\chi_{i1}(t)\varepsilon_{ri})\tau_{ri}(t)+\chi_{i2}(t)p_{ri} \end{array}\right.$$
where $\chi_{i1}(t)$ and $\chi_{i2}(t)$ are time-varying parameters, $\left|\chi_{i1}(t)\right|\leq 1$, and $\left|\chi_{i2}(t)\right|\leq 1$. Therefore, the actual controller (34) can be rewritten as:(39)$$\left\{\begin{array}{l} \tau_{ui}=\frac{h_{ui}(t)}{1+\chi_{i1}(t)\varepsilon_{ui}}-\frac{\chi_{i2}(t)p_{ui}}{1+\chi_{i1}(t)\varepsilon_{ui}}\\ \tau_{ri}=\frac{h_{ri}(t)}{1+\chi_{i1}(t)\varepsilon_{ri}}-\frac{\chi_{i2}(t)p_{ri}}{1+\chi_{i1}(t)\varepsilon_{ri}} \end{array}\right.$$

Thus, substituting (39) into (37) yields:(40)$$\begin{array}{ll} {\dot{V}}_{i3}= & -k_{di}{\left(\frac{L_{\varpi i}e_{\varpi i}}{L_{\varpi i}-e_{\varpi i}}\right)}^{2}-L_{di}{\left(\frac{L_{\varpi i}e_{\varpi i}}{L_{\varpi i}-e_{\varpi i}}\right)}^{3/2}-\frac{e_{f,ui}^{2}}{\xi_{ui}}-2e_{ui}e_{\varpi i}s_{\varpi i}\cos\phi_{i}-e_{f,ui}N_{ui}(\cdot )+e_{ui}f_{i1}+\frac{e_{f,ui}e_{ui}}{\xi_{ui}}\\ & -k_{ai}{\left(\frac{L_{\phi i}e_{\phi i}}{L_{\phi i}-e_{\phi i}}\right)}^{2}-L_{ai}{\left(\frac{L_{\phi i}e_{\phi i}}{L_{\phi i}-e_{\phi i}}\right)}^{3/2}-\frac{e_{f,ri}^{2}}{\xi_{ri}}-2e_{ri}e_{\phi i}s_{\phi i}-e_{f,ri}N_{ri}(\cdot )+e_{f,ri}f_{i3}+\frac{e_{f,ri}e_{ri}}{\xi_{ri}}\\ & +\frac{m_{i,22}e_{ri}}{{\bar{m}}_{i,33}}\left(\frac{h_{ri}(t)}{1+\chi_{i1}(t)\varepsilon_{ri}}-\frac{\chi_{i2}(t)p_{ri}}{1+\chi_{i1}(t)\varepsilon_{ri}}\right)+\frac{e_{ui}}{m_{i,11}}\left(\frac{h_{ui}(t)}{1+\chi_{i1}(t)\varepsilon_{ui}}-\frac{\chi_{i2}(t)p_{ui}}{1+\chi_{i1}(t)\varepsilon_{ui}}\right) \end{array}$$

From Lemma 2, because a∈ℝ and $\varepsilon_{0}>0$, $-a\tanh\left(\frac{a}{\varepsilon_{0}}\right)\leq 0$; we can obtain $e_{ui}h_{ui}\leq 0$ and $e_{ri}h_{ri}\leq 0$ from (33). For $\left|\chi_{i1}(t)\right|\leq 1$ and $\left|\chi_{i2}(t)\right|\leq 1$, satisfy:(41)$$\left\{\begin{array}{l} \frac{e_{ui}h_{ui}(t)}{1+\chi_{i1}(t)\varepsilon_{ui}}\leq \frac{e_{ui}h_{ui}(t)}{1+\varepsilon_{ui}}\\ \frac{e_{ri}h_{ri}(t)}{1+\chi_{i1}(t)\varepsilon_{ri}}\leq \frac{e_{ri}h_{ri}(t)}{1+\varepsilon_{ri}}\\ -\frac{e_{ui}\chi_{i2}(t)p_{ui}}{1+\chi_{i1}(t)\varepsilon_{ui}}\leq \left|\frac{e_{ui}p_{ui}}{1-\varepsilon_{ui}}\right|\\ -\frac{e_{ri}\chi_{i2}(t)p_{ri}}{1+\chi_{i1}(t)\varepsilon_{ri}}\leq \left|\frac{e_{ri}p_{ri}}{1-\varepsilon_{ri}}\right| \end{array}\right.$$

According to Lemma 2, substituting (33) and (41) into (40) yields:(42)$$\begin{array}{ll} {\dot{V}}_{i3}= & -k_{di}{\left(\frac{L_{\varpi i}e_{\varpi i}}{L_{\varpi i}-e_{\varpi i}}\right)}^{2}-L_{di}{\left(\frac{L_{\varpi i}e_{\varpi i}}{L_{\varpi i}-e_{\varpi i}}\right)}^{3/2}-\frac{e_{f,ui}^{2}}{\xi_{ui}}-k_{ui}e_{ui}^{2}-l_{ui}e_{ui}^{3/2}-L_{ui}e_{f,ui}^{3/2}-\left|e_{ui}{\bar{p}}_{ui}\right|+\left|\frac{e_{ui}p_{ui}}{1-\varepsilon_{ui}}\right|-e_{f,ui}N_{ui}(\cdot )\\ & -k_{ai}{\left(\frac{L_{\phi i}e_{\phi i}}{L_{\phi i}-e_{\phi i}}\right)}^{2}-L_{ai}{\left(\frac{L_{\phi i}e_{\phi i}}{L_{\phi i}-e_{\phi i}}\right)}^{3/2}-\frac{e_{f,ri}^{2}}{\xi_{ri}}-k_{ri}e_{ri}^{2}-l_{ri}e_{ri}^{3/2}-L_{ri}e_{f,ri}^{3/2}-\left|e_{ri}{\bar{p}}_{ri}\right|+\left|\frac{e_{ri}p_{ri}}{1-\varepsilon_{ri}}\right|-e_{f,ri}N_{ri}(\cdot )+1.114\sigma \end{array}$$

Consider the sets $\Omega_{di}≜\left\{{\left\|\eta_{i-1}\right\|}^{2}+{\left\|{\dot{\eta }}_{i-1}\right\|}^{2}+{\left\|{\ddot{\eta }}_{i-1}\right\|}^{2}+{\left\|\upsilon_{i}\right\|}^{2}+{\left\|k_{\varpi i}\right\|}^{2}+{\left\|{\dot{k}}_{\varpi i}\right\|}^{2}+{\left\|{\ddot{k}}_{\varpi i}\right\|}^{2}\leq B_{di}\right\}$ with $B_{di}>0$. Consider the sets $\Omega_{ei}≜\left\{{\left\|e_{\varpi i}\right\|}^{2}+{\left\|e_{\phi i}\right\|}^{2}+{\left\|e_{ui}\right\|}^{2}+{\left\|e_{ri}\right\|}^{2}+{\left\|e_{f,ui}\right\|}^{2}+{\left\|e_{f,ri}\right\|}^{2}\leq B_{ei}\right\}$ with $B_{ei}>0$. $B_{di}$ and $B_{ei}$ are compact sets. From (27), all the error variables in the functions $N_{ui}(\cdot )$ and $N_{ri}(\cdot )$ are bounded in the compact set $\Omega_{di}\times \Omega_{ei}$, and it follows that constants $N_{ui}^{*}$ and $N_{ri}^{*}$ exist with $\left|N_{ui}(\cdot )\right|\leq N_{ui}^{*}$ and $\left|N_{ri}(\cdot )\right|\leq N_{ri}^{*}$.

By completion of squares, the following inequalities hold:(43)$$\left\{\begin{array}{c} -e_{f,ui}N_{ui}(\cdot )\leq \frac{e_{f,ui}^{2}}{2}+\frac{N_{ui}^{*}{}^{2}}{2}\\ -e_{f,ri}N_{ri}(\cdot )\leq \frac{e_{f,ri}^{2}}{2}+\frac{N_{ri}^{*}{}^{2}}{2} \end{array}\right.$$

For $\frac{p_{ui}}{1-\varepsilon_{ui}}<{\bar{p}}_{ui}$ and $\frac{p_{ri}}{1-\varepsilon_{ri}}<{\bar{p}}_{ri}$, substituting these and (43) into (42) yields:(44)$$\begin{array}{ll} {\dot{V}}_{i3}= & -k_{di}{\left(\frac{L_{\varpi i}e_{\varpi i}}{L_{\varpi i}-e_{\varpi i}}\right)}^{2}-L_{di}{\left(\frac{L_{\varpi i}e_{\varpi i}}{L_{\varpi i}-e_{\varpi i}}\right)}^{3/2}-\left(\frac{2-\xi_{ui}}{2\xi_{ui}}\right)e_{f,ui}^{2}-k_{ui}e_{ui}^{2}-l_{ui}e_{ui}^{3/2}-L_{ui}e_{f,ui}^{3/2}+\frac{N_{ui}^{*}{}^{2}}{2}\\ & -k_{ai}{\left(\frac{L_{\phi i}e_{\phi i}}{L_{\phi i}-e_{\phi i}}\right)}^{2}-L_{ai}{\left(\frac{L_{\phi i}e_{\phi i}}{L_{\phi i}-e_{\phi i}}\right)}^{3/2}-\left(\frac{2-\xi_{ri}}{2\xi_{ri}}\right)e_{f,ri}^{2}-k_{ri}e_{ri}^{2}-l_{ri}e_{ri}^{3/2}-L_{ri}e_{f,ri}^{3/2}+\frac{N_{ri}^{*}{}^{2}}{2}+1.114\sigma \end{array}$$
Thus, (44) becomes:(45)$${\dot{V}}_{i3}(x)\leq -\kappa_{i1}V_{i3}-\kappa_{i2}V_{i3}^{3/4}+\varepsilon_{i}^{*}$$
where
(46)$$\begin{array}{l} \kappa_{i1}=\min\left\{\frac{k_{di}}{2},\frac{k_{\phi i}}{2},\left(\frac{2-\xi_{ui}}{\xi_{ui}}\right),\left(\frac{2-\xi_{ri}}{\xi_{ri}}\right),2k_{ui},2k_{ri}\right\}\\ \kappa_{i2}=\min\left\{2^{\frac{3}{4}}L_{di},2^{\frac{3}{4}}L_{ai},2^{\frac{3}{4}}l_{ui},2^{\frac{3}{4}}l_{ri},2^{\frac{3}{4}}L_{ui},2^{\frac{3}{4}}L_{ri}\right\}\\ \varepsilon_{i}^{*}=\frac{N_{ui}^{*}{}^{2}}{2}+\frac{N_{ri}^{*}{}^{2}}{2}+1.114\sigma \end{array}$$

Therefore, all signals of the control system can converge to a circular region $\Omega_{Li}=\min\left\{\frac{\varepsilon_{i}^{*}}{(1-H)\kappa_{i1}},{\left(\frac{\varepsilon_{i}^{*}}{(1-H)\kappa_{i2}}\right)}^{4/3}\right\}$ near the origin in a practical finite time, according to Lemma 1:(47)$$T_{i}\leq \max\left\{\frac{4}{\kappa_{i1}H}\ln\left(\frac{\kappa_{i1}HV^{1/4}(0)+\kappa_{i2}}{\kappa_{i2}}\right),\frac{4}{\kappa_{i1}}\ln\left(\frac{\kappa_{i1}V^{1/4}(0)+\kappa_{i2}H}{\kappa_{i2}H}\right)\right\}$$

By choosing appropriate design parameters $\varepsilon_{ui},\varepsilon_{ri},{\bar{p}}_{ui},{\bar{p}}_{ri},p_{ui},p_{ri},\xi_{ui},\xi_{ri},k_{ui},k_{ri},k_{di},k_{ai},$ and $L_{di},L_{ai},l_{ui},l_{ri},L_{ui},L_{ri}$, $\Omega_{Li}$ values are limited to a region $C_{i}^{*}=\min\left\{\frac{\varepsilon_{i}^{*}}{(1-H)\kappa_{i1}},{\left(\frac{\varepsilon_{i}^{*}}{(1-H)\kappa_{i2}}\right)}^{4/3}\right\}$, with:(48)$$\frac{1}{2}{\left(\frac{L_{\varpi i}e_{\varpi i}}{L_{\varpi i}-e_{\varpi i}}\right)}^{2}+\frac{1}{2}{\left(\frac{L_{\phi i}e_{\phi i}}{L_{\phi i}-e_{\phi i}}\right)}^{2}+\frac{e_{f,ui}^{2}}{2}+\frac{e_{f,ri}^{2}}{2}+\frac{e_{ui}^{2}}{2}+\frac{e_{ri}^{2}}{2}\leq C_{i}^{*}$$
Thus:(49)$$\frac{1}{2}{\left(\frac{L_{\varpi i}e_{\varpi i}}{L_{\varpi i}-e_{\varpi i}}\right)}^{2}\leq C_{i}^{*},-\frac{\sqrt{2C_{i}^{*}}L_{\varpi i}}{\sqrt{2C_{i}^{*}}+L_{\varpi i}}\leq e_{\varpi i}\leq \frac{\sqrt{2C_{i}^{*}}L_{\varpi i}}{\sqrt{2C_{i}^{*}}+L_{\varpi i}},-\frac{L_{\varpi i}\left(\sqrt{2C_{i}^{*}}+L_{\varpi i}\right)}{\sqrt{2C_{i}^{*}}+L_{\varpi i}}<e_{\varpi i}<\frac{L_{\varpi i}\left(\sqrt{2C_{i}^{*}}+L_{\varpi i}\right)}{\sqrt{2C_{i}^{*}}+L_{\varpi i}}$$
(50)$$\frac{1}{2}{\left(\frac{L_{\phi i}e_{\phi i}}{L_{\phi i}-e_{\phi i}}\right)}^{2}\leq C_{i}^{*},-\frac{\sqrt{2C_{i}^{*}}L_{\phi i}}{\sqrt{2C_{i}^{*}}+L_{\phi i}}\leq e_{\phi i}\leq \frac{\sqrt{2C_{i}^{*}}L_{\phi i}}{\sqrt{2C_{i}^{*}}+L_{\phi i}},-\frac{L_{\phi i}\left(\sqrt{2C_{i}^{*}}+L_{\phi i}\right)}{\sqrt{2C_{i}^{*}}+L_{\phi i}}<e_{\phi i}<\frac{L_{\phi i}\left(\sqrt{2C_{i}^{*}}+L_{\phi i}\right)}{\sqrt{2C_{i}^{*}}+L_{\phi i}}$$

From (48), (49) and (50), satisfy the following:(51)$$\left|e_{\varpi i}\right|<L_{\varpi i},\left|e_{\phi i}\right|<L_{\phi i},\left|e_{f,ui}\right|<\sqrt{2C_{i}^{*}},\left|e_{f,ri}\right|<\sqrt{2C_{i}^{*}},\left|e_{ui}\right|<\sqrt{2C_{i}^{*}},\left|e_{ri}\right|<\sqrt{2C_{i}^{*}}$$

In summary, the errors $e_{\varpi i}$, $e_{\phi i}$, $e_{f,ui}$, $e_{f,ri}$, $e_{ui}$ and $e_{ri}$ are stable for a practical finite time, and the follower tracks its leader to complete the specific formation in finite time.

Motivated by [34], if $t_{ui}^{*}>0$ and $t_{ri}^{*}>0$, the lower bound of the trigger interval $t_{k+1}^{ui}-t_{k}^{ui}$ is $t_{ui}^{*}$, and $t_{k+1}^{ri}-t_{k}^{ri}$ is $t_{ri}^{*}$. Therefore, combining $E_{ui}(t)=h_{ui}(t)-\tau_{ui}(t),$$\forall t\in \left[t_{k}^{ui},\right.\left.t_{k+1}^{ui}\right)$ and $E_{ri}(t)=h_{ri}(t)-\tau_{ri}(t),\forall t\in \left[t_{k}^{ri},\right.\left.t_{k+1}^{ri}\right)$, we obtain:(52)$$\left\{\begin{array}{l} \frac{d}{dt}\left|E_{ui}\right|=sign(E_{ui}){\dot{E}}_{ui}\leq \left|{\dot{h}}_{ui}\right|\\ \frac{d}{dt}\left|E_{ri}\right|=sign(E_{ri}){\dot{E}}_{ri}\leq \left|{\dot{h}}_{ri}\right| \end{array}\right.$$

It can be seen from the above discussion that all signals are bounded, i.e., there is $\gamma_{ui}>0$ and $\gamma_{ri}>0$; and $\left|{\dot{h}}_{ui}\right|\leq \gamma_{ui}$ and $\left|{\dot{h}}_{ri}\right|\leq \gamma_{ri}$. In addition, $E_{ui}(t)=0$, $E_{ri}(t)=0$, $\underset{t\to t_{k+1}^{ui}}{\lim}E_{ui}(t)=\varepsilon_{ui}\left|\tau_{ui}(t)\right|+p_{ui}$ and $\underset{t\to t_{k+1}^{ri}}{\lim}E_{ri}(t)=\varepsilon_{ri}\left|\tau_{ri}(t)\right|+p_{ri}$; thus, $t_{ui}^{*}\geq \frac{\varepsilon_{ui}\left|\tau_{ui}(t)\right|+p_{ui}}{\gamma_{ui}}$ and $t_{ri}^{*}\geq \frac{\varepsilon_{ri}\left|\tau_{ri}(t)\right|+p_{ri}}{\gamma_{ri}}$. Therefore, the proposed control system avoids Zeno behavior [35]. The proof is complete. □

## 4. Simulations

In this section, numerical simulations are conducted to verify the effectiveness and tracking performance of the proposed relative threshold-based finite-time event-triggered control method. The virtual leader’s trajectory is set to $h_{0}={\left[100\sin(0.01t),60(1-\cos(0.01t))\right]}^{T}$. The initial states of the ASVs are chosen as $\eta_{1}(0)={\left[-5.1,0,-0.02\right]}^{T}$, $\eta_{2}(0)={\left[-9,-3,0.01\right]}^{T}$ and $\eta_{3}(0)={\left[-13.1,-6,0.02\right]}^{T}$. The parameters of the underactuated ASV dynamic model are shown in Table 1 [31].

The performance functions shown in (16) are chosen as $L_{\varpi i}(t)=(0.5-0.06)\exp(-0.1t)+0.06$ and $L_{\phi i}(t)=(\frac{\pi }{8}-0.06)\exp(-0.1t)+0.06$. The underactuated ASV formation distance and angle constraint parameters are given by ${\underline{\varpi }}_{i}=4.5m$, ${\bar{\varpi }}_{i}=5.5m$, ${\underline{\phi }}_{1}=-\pi /8$, ${\bar{\phi }}_{1}=\pi /8$, ${\underline{\phi }}_{2}={\underline{\phi }}_{3}=\pi /8$ and ${\bar{\phi }}_{2}={\bar{\phi }}_{3}=3\pi /8$. The desired tracking distance and angle are given by $\varpi_{i,des}=5$, $\phi_{1,des}=0$ and $\phi_{2,des}=\pi /4$. Table 2 shows the design parameters.

In Figure 3, each ASV tracks its leader while maintaining the desired distance between a group of leaders and followers. The LOS range and angle error satisfy the prescribed performance specification, realize the connectivity of formation control communication and avoid collision, as shown in Figure 4 and Figure 5. Figure 6, Figure 7 and Figure 8 show the control input with the event-triggering mechanism. In Figure 6, Figure 7 and Figure 8, the control inputs of each ASV are continuous and chattering-free. The control input of ASV1 based on time-triggered development is continuously updated, as shown in Figure 9. The tracking performance of the time-triggered control of ASV1 is shown in Figure 10 and Figure 11. As shown in Figure 9, Figure 10 and Figure 11, compared with time-triggered control, the event-triggered control strategy has better control performance and saves more communication resources. The triggering effect of the proposed event-triggered strategy based on the relative threshold is shown in Table 3 and shows that the proposed event-triggered strategy can save considerable communication bandwidth.

In addition, to verify the performance of the proposed control strategy, it is compared with the tan-type Lyapunov function (TBLF) control strategy in [12]. From Figure 12, compared with the TBLF strategy, this control method has a smaller tracking error and faster convergence rate. The maximum steady-state error of the proposed control method does not exceed 0.005, but the maximum of the TBLF strategy can reach approximately 0.02. As shown in Figure 13 and Figure 14, when the control input is basically the same, the angle tracking error of the control algorithm in this paper is less than that of TBLF method. In summary, the control system designed in this paper has good control performance and saves considerable communication resources.

Compared with the analysis of the fixed-threshold event-triggered strategy (FET) [26], the event-triggered strategy based on the relative threshold proposed in this paper can greatly reduce the communication resources used by the system. As shown in Figure 15 and Figure 16, the proposed event-triggered strategy based on the relative threshold has a faster response speed and better tracking accuracy. As shown in Table 3, using the FET method, the percentages of ASV1 communication triggers in the total communication time are changed from 4.44% and 3.52% to 4.51% and 4.07%, respectively, which indicates that the proposed strategy can save more communication resources than the FET method.

## 5. Conclusions

In this paper, a finite-time event-triggered formation controller is proposed to solve the problem of achieving underactuated ASV formation control with limited communication resources and limited performance while maintaining communication efficiency and avoiding collisions. By developing an improved BLF, prescribed transient tracking error and steady-state performance are guaranteed, and the maintenance of communication distance and collision avoidance between the ASV leader and the follower is realized. To reduce the communication bandwidth, a relative threshold event-triggered strategy is proposed. The results of stability analysis show that all signals of the control system are PFS. Finally, the simulation results show that the proposed control method is effective and feasible. Typically, actuator output limitations have a significant impact on ASVs, and when actuator output saturation occurs, it may have some impact on the control accuracy of the control system. Moreover, actuator faults resulting in insufficient actuator output may not only affect the control system accuracy but also lead to control system paralysis. In practice, the impact of the environmental load cannot be ignored in the navigation mission of ASVs. Therefore, in future work, the environmental load, actuator output saturation and actuator faults will be considered, and the proposed control method will be applied to practical scenarios to further improve the experimental verification and optimize the control system.

## Figures and Tables

**Figure 1 sensors-23-06756-f001:**
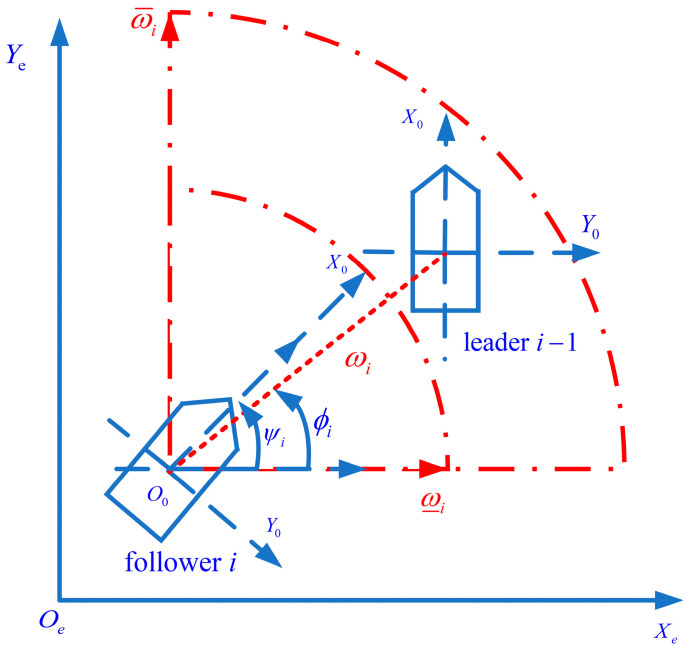
The architecture of a group of leader-followers.

**Figure 2 sensors-23-06756-f002:**
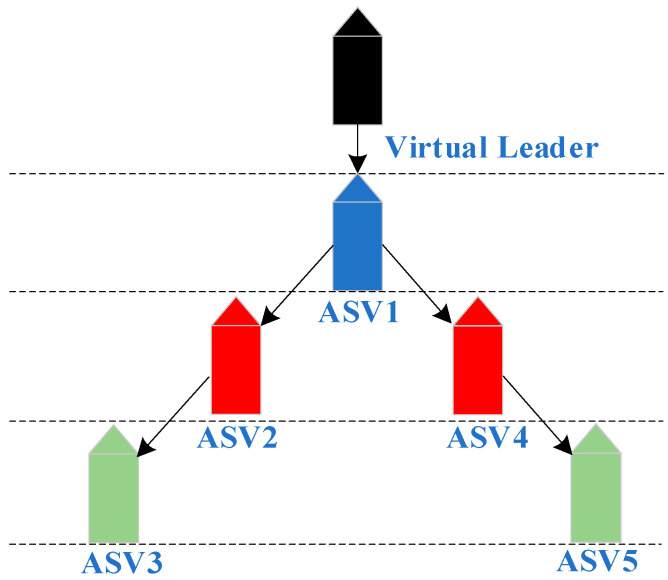
Formation structure of underactuated ASVs.

**Figure 3 sensors-23-06756-f003:**
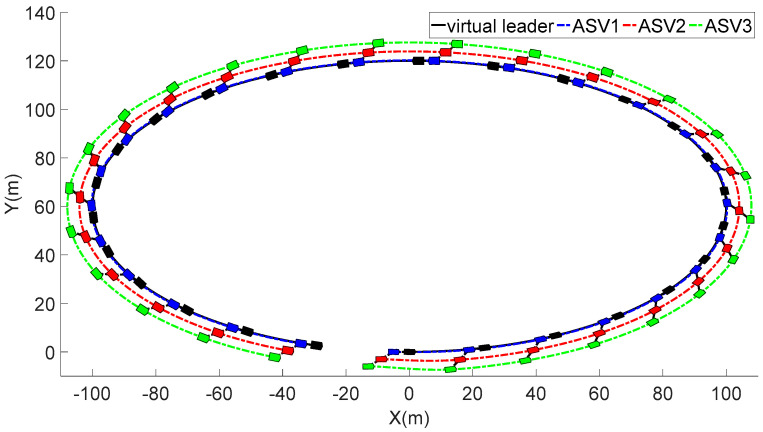
The formation control trajectory.

**Figure 4 sensors-23-06756-f004:**
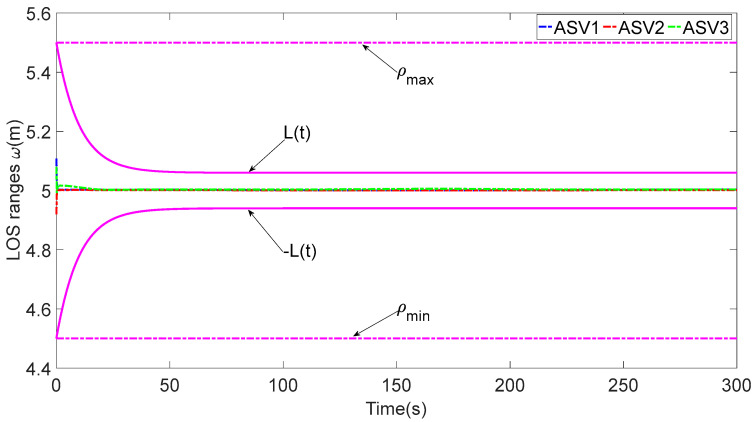
The LOS range $\omega_{i}$.

**Figure 5 sensors-23-06756-f005:**
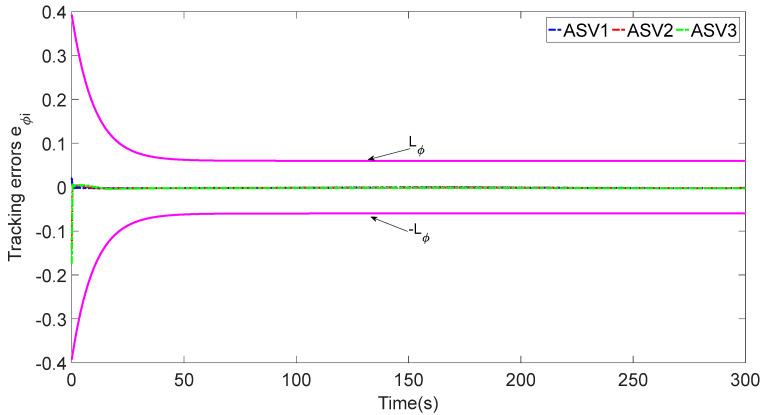
The tracking errors $e_{\phi i}$.

**Figure 6 sensors-23-06756-f006:**
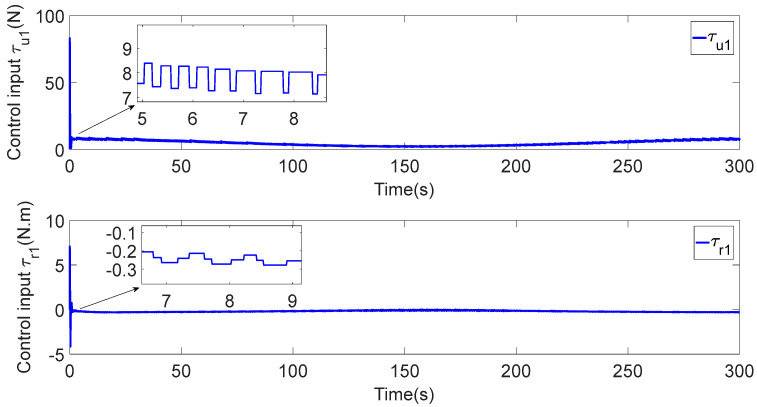
Control input of ASV1.

**Figure 7 sensors-23-06756-f007:**
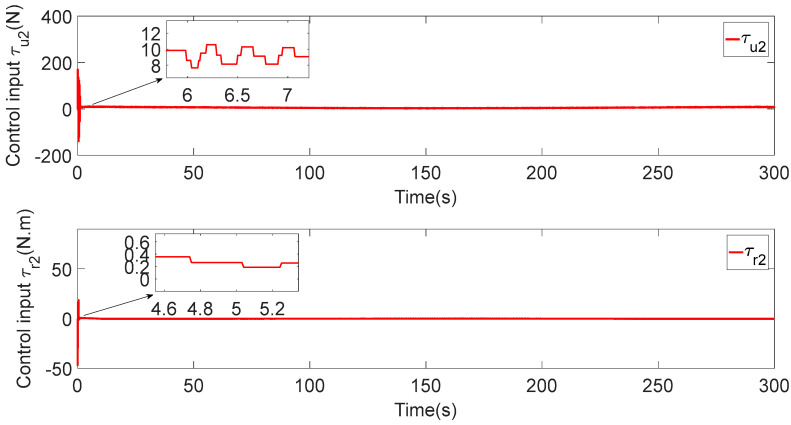
Control input of ASV2.

**Figure 8 sensors-23-06756-f008:**
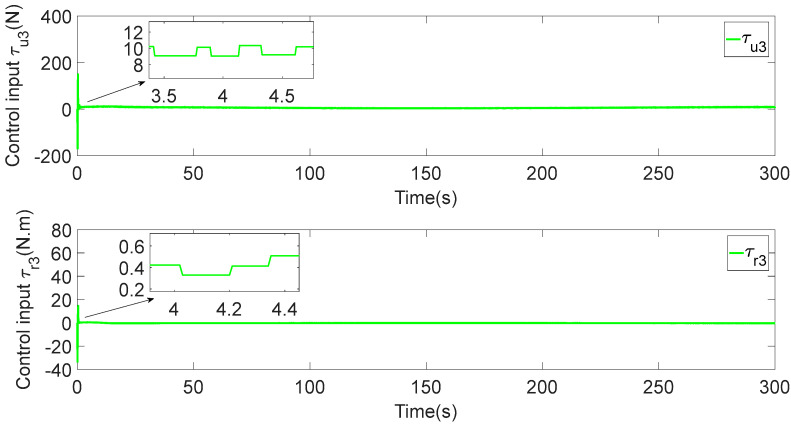
Control input of ASV3.

**Figure 9 sensors-23-06756-f009:**
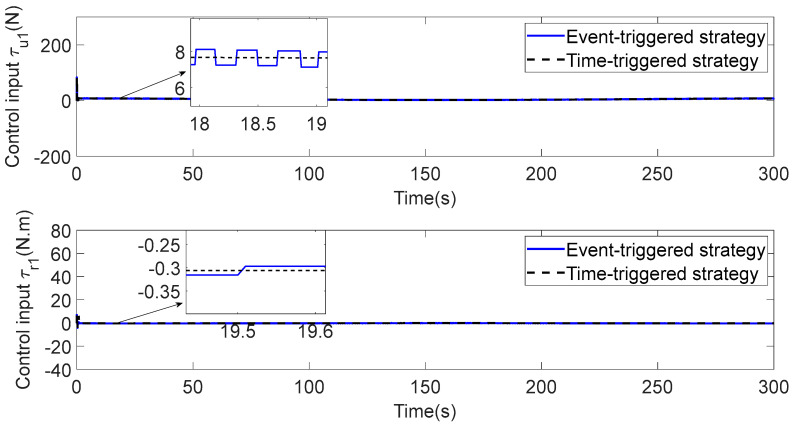
Comparison of control input (ASV1).

**Figure 10 sensors-23-06756-f010:**
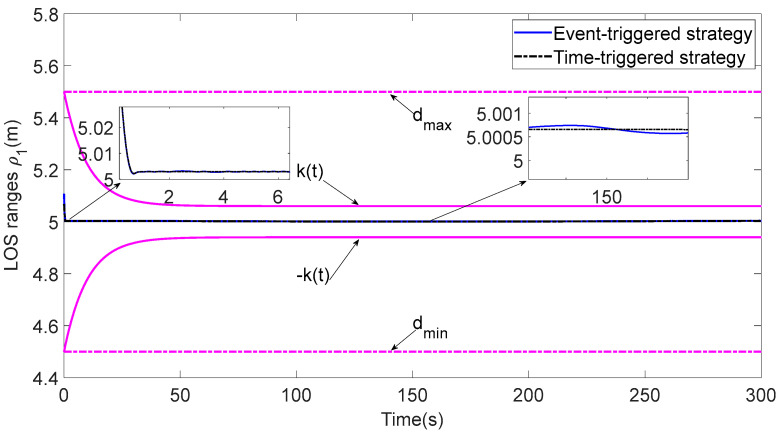
Comparison of the LOS range.

**Figure 11 sensors-23-06756-f011:**
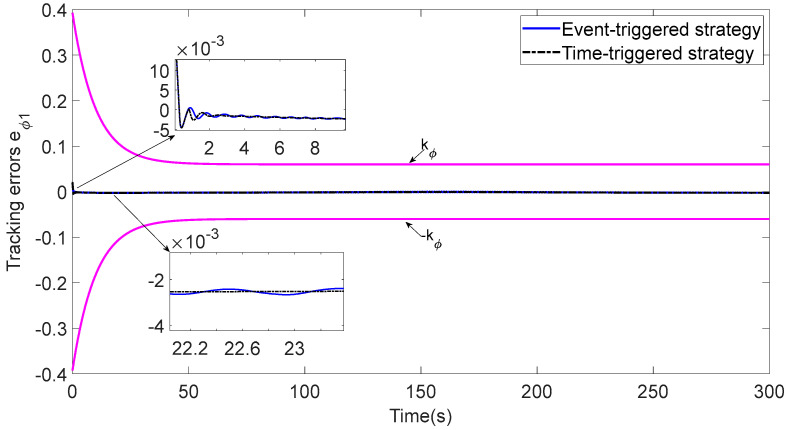
Comparison of tracking errors.

**Figure 12 sensors-23-06756-f012:**
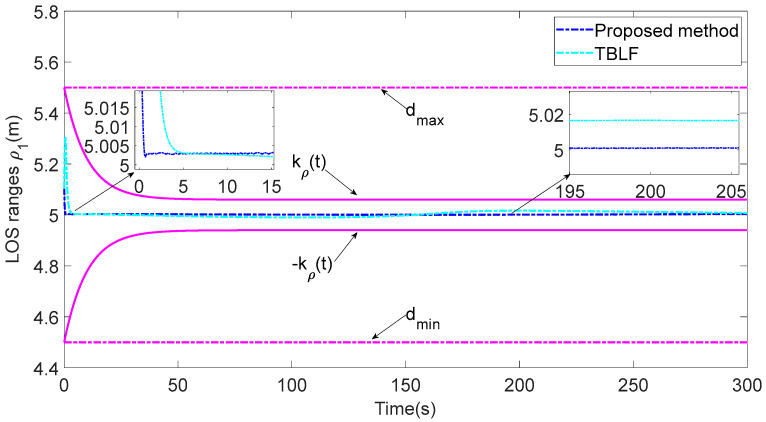
Comparisons of the LOS range $\rho_{1}$.

**Figure 13 sensors-23-06756-f013:**
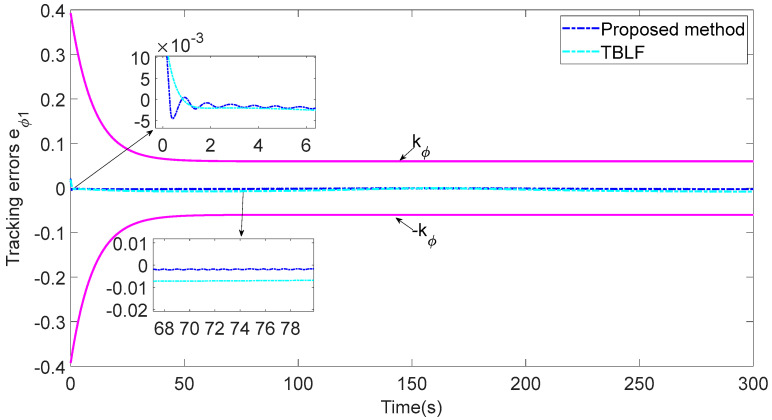
Comparisons of the tracking error $e_{\phi 1}$.

**Figure 14 sensors-23-06756-f014:**
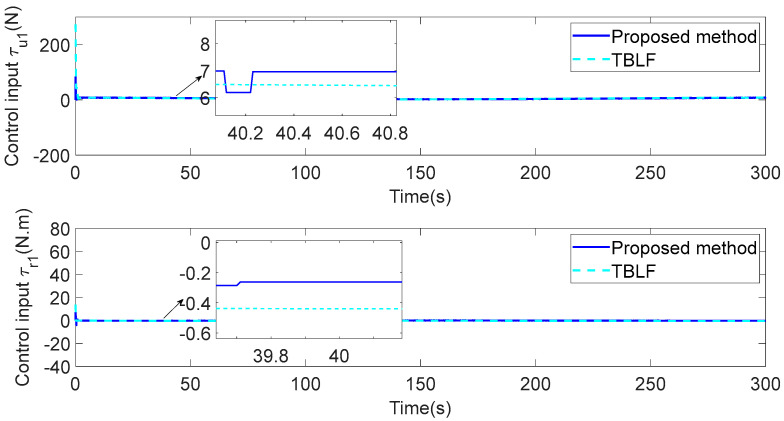
Comparisons of the control inputs.

**Figure 15 sensors-23-06756-f015:**
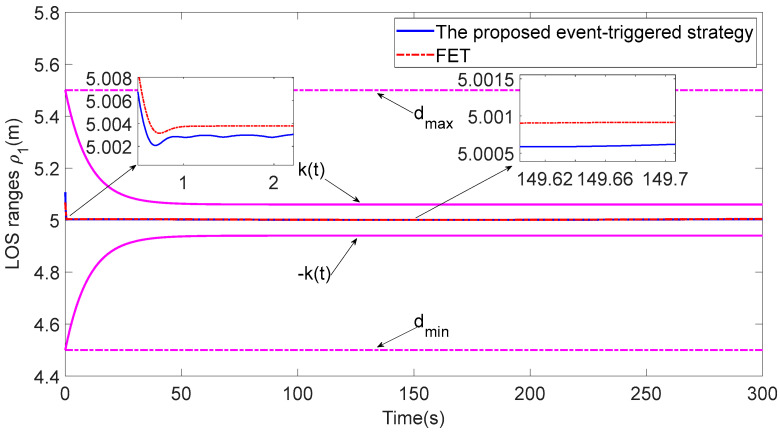
Comparison of the LOS range with FET.

**Figure 16 sensors-23-06756-f016:**
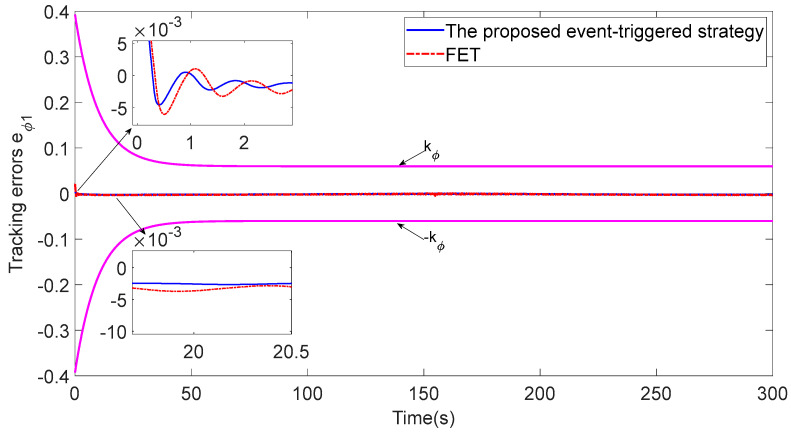
Comparisons of the tracking error $e_{\phi 1}$ with FET.

**Table 1 sensors-23-06756-t001:** The model parameters of the underactuated ASV.

Parameter	Value
$m_{i,11}$	25.8000
$m_{i,22}$	33.8000
$m_{i,23}=m_{i,32}$	1.0115
$m_{i,33}$	2.7600
$C_{i,13}=-C_{i,31}$	$-1.0115r_{i}-33.8000v_{i}$
$C_{i,23}=-C_{i,32}$	$25.8000u_{i}$
$d_{i,11}$	$5.8664u_{i}^{2}+1.3274\left|u_{i}\right|+0.7225$
$d_{i,22}$	$0.8050\left|r_{i}\right|+36.2823\left|v_{i}\right|+0.8612$
$d_{i,23}$	$0.8450\left|v_{i}\right|+3.4500\left|r_{i}\right|-0.1079$
$d_{i,32}$	$-5.0437\left|v_{i}\right|-0.1300\left|r_{i}\right|-0.1025$
$d_{i,33}$	$0.7500\left|r_{i}\right|-0.0800\left|v_{i}\right|+1.9000$

**Table 2 sensors-23-06756-t002:** Control parameters.

Parameter	Value	Parameter	Value	Parameter	Value
$\varepsilon_{ui}$	0.10	$k_{ai}$	2	$p_{ui}$	0.05
$\varepsilon_{ri}$	0.10	$L_{di}$	1	$p_{ri}$	0.05
${\bar{p}}_{ui}$	0.06	$L_{ai}$	1	$\xi_{ui}$	0.01
${\bar{p}}_{ri}$	0.06	$l_{ui}$	5	$\xi_{ri}$	0.01
$k_{ui}$	2.00	$l_{ri}$	7.00		
$k_{ri}$	4.00	$L_{ui}$	5.00		
$k_{di}$	1.00	$L_{ri}$	7.00		

**Table 3 sensors-23-06756-t003:** Number of communication triggers and percentage of total communication time (total sampled data: 60,000).

Variable	Triggering Time	Percentage
$\tau_{u1}$	2665	4.44%
$\tau_{r1}$	2110	3.52%
$\tau_{u2}$	6272	10.45%
$\tau_{r2}$	2134	3.56%
$\tau_{u3}$	4158	6.93%
$\tau_{r3}$	1857	3.09%
FET-$\tau_{u1}$	2707	4.51%
FET-$\tau_{r1}$	2445	4.07%

## Data Availability

Not applicable.

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
