# Peer review of "Event-Triggered Finite-Time Formation Control of Underactuated Multiple ASVs with Prescribed Performance and Collision Avoidance"

_sensors, 2023, doi:10.3390/s23156756_

Round 1

Reviewer 2 Report

This paper presented the “Event-triggered finite-time formation control of underactuated multiple ASVs with prescribed performance and collision avoidance”. This work aims to solve the problem of underactuated multiple autonomous surface vessels, including systematic mass matrix, collision avoidance, maintaining communication distances and prescribed performance. The results provide a comprehensive finite-time formation control method to avoid collisions and maintain communication distance. The contribution can be applied to traffic management in the maritime industry. This paper is relevant and valuable to the readers of Sensors. However, several places need to be revised. There, I recommend publication of this paper subject after major revision. The following changes to improve this paper was suggested.

  1. Introduction, the authors need to make a simple definition for “ASVs” because it is the keyword in this work. Also, please give some simple samples to explain why we need to develop ASVs technology.
  2. Please organize the structure for this work at the end of the Introduction. We need more information before reading this reading work.
  3. Although the authors point out the contribution of this work, however, we need to know the difference between this work and others. Please explain what is revolution or brilliant can beyond previous work.
  4. Page 12, line no. 328 to 337, there are some “errors” in your paper. Please check them before submitting them.
  5. Please present the work environment for your simulation when ship operations at sea or port have many complicated settings; I feel this test could be more workable. Can they be actual employ in ship operations? How about the ship size? Not only figures simulation but also need to present the environment.

Non.

Round 2

Reviewer 2 Report

This paper presented the “Event-triggered finite-time formation control of underactuated multiple ASVs with prescribed performance and collision avoidance”. This work aims to solve the problem of underactuated multiple autonomous surface vessels, including systematic mass matrix, collision avoidance, maintaining communication distances and prescribed performance. The results provide a comprehensive finite-time formation control method to avoid collisions and maintain communication distance. The contribution can be applied to traffic management in the maritime industry. This paper is relevant and valuable to the readers of Sensors.

 The authors have modified all the problems at my request. I have no further problems.

Thank you.

Non.